# Challenges in Manufacturing of Hemp Fiber-Reinforced Organo Sheets with a Recycled PLA Matrix

**DOI:** 10.3390/polym15224357

**Published:** 2023-11-08

**Authors:** Maximilian Salmins, Florian Gortner, Peter Mitschang

**Affiliations:** Leibniz-Institut für Verbundwerkstoffe GmbH, Erwin-Schroedinger-Strasse 58, 67663 Kaiserslautern, Germany

**Keywords:** natural fiber reinforced thermoplastic polymers, organo sheet, recycled PLA, hemp fiber, B-factor model

## Abstract

This study investigates the influence of a hot press process on the properties of hemp fiber-reinforced organo sheets. Plain-woven fabric made from hemp staple fiber yarns is used as textile reinforcement, together with a recycled poly-lactic acid (PLA) matrix. Process pressure and temperature are considered with three factor levels for each parameter. The parameter influence is examined based on the B-factor model, which considers the temperature-dependent viscosity of the polymer, as well as the process pressure for the calculation of a dimensionless value. Increasing these parameters theoretically promotes improvements in impregnation. This study found that the considered recycled polymer only allows a narrow corridor to achieve adequate impregnation quality alongside optimal bending properties. Temperatures below 170 °C impede impregnation due to the high melt viscosity, while temperature increases to 185 °C show the first signs of thermal degradation, with reduced bending modulus and strength. A comparison with hemp fiber-reinforced virgin polypropylene, manufactured with identical process parameters, showed that this reduction can be mainly attributed to polymer degradation rather than reduction in fiber properties. The process pressure should be at least 1.5 MPa to allow for sufficient compaction of the textile stack, thus reducing theoretical pore volume content to a minimum.

## 1. Introduction

Global challenges such as the scarcity of fossil resources demand increasing efforts to substitute these materials. The substitution can be achieved by using bio-based materials or by recycling instead of using virgin materials [1,2,3]. The use of bio-based materials can also lead to reduced energy consumption, compared to conventional materials, e.g., glass fiber-reinforced polymer composites [4,5,6]. Natural fiber-reinforced polymer composites are attractive materials for use in automotive applications—for which up to 37% of total biocomposites in Europe are produced [7]. Hemp fibers are bast fibers and are among the most commonly used natural fibers for composite applications [4,7,8,9,10]. They can be cultivated in Europe, which further enables sustainable manufacturing with limited transport distance, compared to natural fibers cultivated in southeast Asia [11,12]. Bast fibers have a particular structure, as depicted in Figure 1. Hemp plants cultivated in Europe reach heights of about 2 m with stem diameters between 3.5 and 8.4 cm. Hemp plants cultivated in China can grow to heights between 4 m and 7 m [8]. The stem consists of the outermost layer, the epidermis, which surrounds several fiber bundles. Those can be up to 3 m long but are cut down to a length of 1 m before processing. The fiber bundles are composed of elementary fibers—often between two and twenty, but sometimes up to forty elementary fibers—which are held together by pectin. The length of the elementary fibers can be up to 100 mm, with the most common range being reported between 10 mm and 40 mm [13]. Typical fiber diameters vary between 15 µm and 50 µm. The center of the stem can be described as a woody core followed by the innermost part, the lumen, which is hollow. This hollow lumen can also be found inside elementary fibers [8].

The final morphology, the degree of fiber separation and the fiber properties depend on a multitude of conditions during cultivation and retting [10,13,14,15]. The extraction of primary or elementary fibers from the plant stem begins with the retting process, in which the surrounding tissue is dissolved by bacteria and fungi [14,16]. Field retting is the most common process. Plant stems remain on the field for several weeks and plant tissue is dissolved by microbiological organisms. This process has to be monitored to prevent overretting and decreases in mechanical properties [14,15]. However, some fibers can still be collected prematurely, which in turn leads to incompletely separated fiber bundles. The chemical composition and the resulting fiber properties reported in the literature may vary widely due to these often uncontrollable influences. The fiber bundles are separated from wooden parts by mechanical breaking and scutching processes after the retting [1,8,11,12,14,17].

Duval et al. [13] studied the tensile properties of hemp fibers to compare the impact of the sampling area within the stem—either the top, middle or bottom. They found limited variations in mechanical properties between the sampling areas. This is mainly due to the standard deviation, which did not allow significant statistical differentiation. However, they described that fiber taken from the middle of the stem showed the highest tensile strengths. They also report that hemp fiber diameter may influence the Young’s modulus—with a hyperbolic decrease with an increasing diameter—while the strength at break showed a higher scatter [13].

Li et al. [18] investigated mechanical properties based on nano-indentation and described similar findings, even though they too were not able to find statistical significance. Placet et al. [19] used a modelling approach to describe the diameter dependence of the Young’s modulus in hemp fibers, in comparison to experimentally determined values. The results for ultimate tensile strength show a hyperbolic dependence with regard to the fiber diameter near the rupture zone. It has also been indicated that clamping distance can have an effect on ultimate tensile strength [10].

The fibers consist of four macromolecules; namely, cellulose, hemicellulose, lignin and pectin. Cellulose functions as the fiber backbone and is thermally stable, withstanding temperatures of up to 350 °C before degradation. Hemicellulose acts as a support material within cell walls. Lignin is a complex molecule that binds cells, fibers and vessels within plants. Both hemicellulose and lignin are reported to show the first signs of thermal degradation at temperatures greater than 200 °C, which overall limits the use of thermoplastic polymers for the manufacturing of composites to those with melt temperatures below this threshold [8].

Polyethylene (PE) [20], polypropylene (PP) [21,22,23,24,25,26,27,28,29] and poly-lactic acid (PLA) [30,31,32] are, therefore, among the most commonly used thermoplastic polymers [33]. Composites with a random fiber orientation made in injection molding processes [4,20,21,22,30,31] or based on non woven fiber mats [4,8,23,24,34] are the biggest share. However, some authors reported approaches for composites with increased mechanical potential due to unidirectional reinforcement [4,27,32,35]. It has been indicated that the mechanical performance of natural fiber-reinforced composites might be limited, due to poor fiber–matrix adhesion, which could be overcome by fiber treatment or matrix functionalization [36,37,38]. PLA is one of the most promising bio-based polymers and has the highest volume of industrial applications [5,7,39,40]. Its comparatively low melting temperature enables the production of natural fiber reinforced composites without compromising the fiber properties [41]. While natural fiber components can withstand temperatures up to 200 °C [8], thermal degradation in PLA is a complex process [42], with some literature indicating a limited thermal stability for temperatures above 190 °C [43]. This degradation is related to the thermal instability of ester groups within the PLA [44], which is further enhanced by moisture, oxygen or mechanical loads [45]. The presence of oxygen may lead to an earlier onset of degradation at 150 °C, compared to 270 °C in vacuum [44]. This behavior may be disadvantageous when the processing of polymers with melt temperatures at these temperatures is considered. Processing PLA at temperatures of 200 °C in air may lead to thermal oxidation with degradation, due to random chain scission [46,47,48].

Continuous reinforcement with natural fibers is possible, when the fibers of a defined length are spun into a staple fiber yarn. Most often, a low fraction of synthetic fibers is used to stabilize the yarn and allow sufficient tensile strengths for weaving processes. These yarns can be used for textile production—i.e., woven or non-crimp fabrics—as a semi-finished product for composite production [9].

Woven fabrics can be used for the production of continuously fiber-reinforced composites, so-called organo sheets. Fiber bundles within the textiles have to be impregnated with the matrix polymer during organo sheet manufacturing. The impregnation of dry fiber textiles with a thermoplastic matrix is a complex process that is influenced by a variety of factors. Figure 2 shows a schematic of the impregnation of a dry textile with a thermoplastic polymer matrix. Both the polymer and the textile are separate, with defined thicknesses X_f(t<0)_, before the process starts. Then, pressure is applied, through which the textile layers are compacted to a defined thickness X_f(t=0)_. The stack is heated to temperatures above the melting temperature of the polymer, which in turn allows the impregnation of the porous textile stack and the displacement of air. The polymer is distributed homogeneously across the textile surface, which facilitates impregnation only through the stack thickness. Displacement of the trapped air allows a hydrostatic equilibrium to be established. This facilitates relaxation of the textile, and its thickness increases again [49,50,51].

The process of impregnation can be divided into two distinct phases—macro- and micro-impregnation. A schematic for material and pore distribution in these phases is shown in Figure 3. The molten polymer impregnates the macro-pores around the fiber bundles during macro-impregnation, whereas the polymer fills the micro-pores within a fiber bundle during micro-impregnation. Both of these processes are influenced by the local permeability. Values of permeability for inter-bundle impregnation—around the fiber bundles—are significantly higher than the values measured for intra-bundle—in between fiber bundles and around individual fibers. High values of permeability facilitate impregnation [52].

The B-factor model, developed by Mayer [53] and refined by Christmann [51], has been proposed as a means to compare the impregnation performance of processes with different parameter combinations. The B-factor model was originally developed in order to allow the transfer of the optimum impregnation parameters determined in a laboratory test to a series production line. The B-factor, as a dimensionless constant, summarizes process parameters such as pressure p and time t, as well as the temperature dependent viscosity *η(T(t))* [54]. The first part of the B-factor is the b-integral, described in Equation (1):(1)$$b=\int_{t_{0}}^{t_{a}}\frac{1}{\eta (T\left(t\right))}dt,$$

This integral can alternatively be calculated as the sum of the individual reciprocal viscosities at time intervals of discrete temperatures, as in Equation (2):(2)$$b=\sum_{k=0}^{n}\left(t_{k}\times \frac{1}{\eta \left(T_{k}\right)}\right),$$

The particular temperature spectrum, in which this value of b-integral is calculated, starts with the point in time at which the melt temperature *T_m_* is exceeded. Usually, the end point for the calculation of *b* is reached once the temperature drops below the crystallization temperature *T_c_* [55]. The temperature-dependent viscosity in this timeframe can be approximated based on an Arrhenius equation, incorporating the viscosity *η*_0_*(T*_0_*)* at a particular temperature, the activation energy *E_a_*, the gas constant *R* and the reciprocal temperature difference, as in Equation (3):(3)$$\eta_{0}\left(T\right)=\eta_{0}\left(T_{0}\right)\times \exp\left[\frac{E_{a}}{R}\times \left(\frac{1}{T}-\frac{1}{T_{0}}\right)\right]$$

The B-factor is then calculated by multiplying this b-integral with the process pressure *p*, as described in Equation (4):(4)$$B=p\times b=p\times \int_{t_{0}}^{t_{a}}\frac{1}{\eta (T\left(t\right))}dt,$$

Previous works reported in the literature mainly focused on the production of natural fiber-reinforced composite parts with a random fiber orientation. However, the possibility to manufacture staple fiber yarns from natural fibers allows for continuous reinforcement, based on woven and non-crimp fabrics. The applicability of the B-factor model to assess impregnation performance has mainly been researched with glass fiber textiles and petrol-based polymers. It should be investigated whether this model is also applicable for the manufacturing of composites with bio-based components—natural fibers or bio-based polymers, or a combination of both. Therefore, this study focusses on the manufacturing of natural fiber-reinforced organo sheets with a fabric reinforcement and a recycled poly-lactic acid matrix polymer, as well as the assessment of the manufacturing process based on the B-factor model.

## 2. Materials and Methods

### 2.1. Materials

Plain woven fabrics with a surface weight of 465 g/m^2^, made from hemp fibers, were used for manufacturing hemp–rPLA organo sheets. The hemp fabric, with a count of 4.9 threads per cm in both warp and weft direction, was made by Gerster TechTex, a division of Gustav Gerster GmbH & Co. KG (located in Biberach an der Riß, Germany), with 400 tex hemp staple fiber yarns produced by Wagenfelder Spinnereien GmbH (located in Wagenfeld, Germany), based on a design developed within the “Durobast” project (www.durobast.de (accessed on 29 September 2023)). A recycled PLA powder from Looplife Polymers was used as the matrix polymer. Borealis Bj100hp polypropylene was used for manufacturing NF–PP organo sheets, in order to identify possible thermal degradation of the natural fibers. Selected polymer properties are given in Table 1. The materials, as well as a schematic for the stacking of the textile and polymer, are shown in Figure 4. The red arrows describe the direction in which the molten polymer flows for fiber impregnation.

### 2.2. Organo Sheet Manufacturing

Organo sheets were manufactured on a laboratory hot press in a modular cylindrical tool. The tool consists of an outer steel ring and two cylindrical steel stamps with a diameter of 100 mm, and each stamp has a thickness of 15 mm. Figure 5 shows the laboratory hot press with the modular cylindrical press tool on the left, as well as the setup for temperature measurement inside the textile stack on the right—each without the outer ring. The stack of textile and polymer was put in between the tool stamps. The stack consisted of three textile layers and 9.3 g of polymer and was prepared to achieve a theoretical thickness of 1.9 mm and a theoretical fiber volume content of 49%. The polymer was put on top of the bottom and middle layers—in this manner, the polymer has to penetrate the outer layers to reach the surface. This setup allowed for instant evaluation of impregnation quality after the manufacturing process. The warp direction within the textile layers was highlighted, before cutting individual pieces on a hydraulic punching unit. The warp direction in each textile layer was aligned within the tool. The hydrophilic behavior and the significant moisture absorption ability of natural fibers are well reported [8,56]. The textile layers and the polymer were therefore dried in a convection oven at 80 °C for a duration of 3 h, to reduce the moisture content and minimize its influence, after which a weight balance was achieved. The average moisture absorption determined for the natural fibers by this drying process was 5%. The pre-dried textiles, together with the polymer powder, were weighed before insertion into the press tool. The organo sheet was again weighed after the process to account for polymer squeeze-out. Organo sheet thickness was measured at four points across the surface, for calculating the theoretical fiber volume content.

The laboratory hot press was designed at Leibniz-Institut für Verbundwerkstoffe and consists of:an electromechanical cylinder for applying forces up to 20 kN in tandem with a membrane load cell as a means of force control,two fixed tool halves for active heating—each tool half with six cartridge heaters with a total output of 2.4 kW, together with sheath thermocouples for temperature control, as well as an active cooling unit with a water–air mixture.

The hot press is controlled by a LabView-based program and is set up for inline recording of process data at 1 Hz, including:temperatures at different locations within the setup (sheath thermocouple within the tool with cartridge heater for temperature control and thermocouples for independent recording; accuracy ± 1 K),effective forces (membrane load cell; accuracy ± 1 N), andthe stack thickness (high accuracy glass scale; accuracy ± 1 µm).

The main test series to identify parameter influences was set up to manufacture hemp–rPLA organo sheets with a target fiber volume content (FVC) of 49%. An overview of the process parameters is presented in Table 2. This test series aimed to identify the optimal process parameters for hemp–rPLA organo sheet manufacturing. An additional test series, with a reduced target FVC of 37%, was conducted with the optimal parameter combination, to compare the effect of FVC on mechanical properties. A set of reference organo sheets with a PP matrix (Borealis bj100hp) was manufactured with a comparable FVC, to investigate the effect of process temperature on bending properties and to identify possible thermal degradation within the hemp fibers. Five organo sheets were manufactured per parameter combination for an adequate statistical basis. The overall timespan of 1280 s for the hot press process was identical for each parameter combination. Process pressure was applied throughout the whole process. The preset variothermal temperature profile was divided into the following phases: 300 s were preset for heating, followed by 600 s of hold time and the final cooling time of 380 s. The effective temperature curve shows a considerable deviation during heating when compared to the preset ramp, which is why the specific process data have to be considered for the evaluation of process impact and impregnation performance. The delay between preset and effective temperature curve has been illustrated by green arrows in Figure 6. The specific thermal expansion of the tool stamps were recorded for each parameter combination. Measurements of the effective temperature inside the tool were conducted without the outer ring, to allow for the placement of thermocouples. A comparison of preset temperature and pressure ramps for organo sheet manufacturing is shown in Figure 6.

### 2.3. Interpretation of Process Data for B-Factor Calculation

Figure 7 shows an exemplary set of process data for effective pressure and stack thickness, together with the temperature curves. This correlation allows the identification of characteristic moments during organo sheet manufacturing which in turn allow dividing the impregnation process into four sections (I–IV). The first section (I) can be described as the stack being fully compacted at moderate temperatures. The following section (II) begins with the temperature exceeding the glass transition temperature, allowing the polymer powder to shift within the stack and providing conditions for progressive compaction and nesting. The third section (III) consists of the to impregnation processes. Macro-impregnation starts with the temperature surpassing the polymer melt temperature. The onset of macro-impregnation can be identified by a distinct drop in process pressure, before a plateau is reached with progresses in micro-impregnation. Macro-impregnation is followed by micro-impregnation, with a distinct plateau in stack thickness. Minimal decreases during this phase can be mainly attributed to polymer squeeze-out through the gap between the tool stamps and the outer ring. The final section (IV) shows decreases in stack thickness, due to solidification during cooling of the setup.

Calculation of the B-factor is based on the reciprocal temperature-specific viscosity and usually begins at the melt temperature of the polymer. This point can be identified by the distinct drop in process pressure for temperatures higher than T_m_ and was identified for each manufacturing process to calculate individual values for the B-factor. The range for calculation usually extends over the time during which the polymer is molten and until the point when organo sheet is cooled below the recrystallization temperature. The considered rPLA does not show a distinct recrystallization behavior during cooling. The range for B-factor calculation within this study is defined as starting with the onset of macro-impregnation, until the point at which the temperature drops below the melt temperature. This range is highlighted in Figure 7.

### 2.4. Consideration of Physical Organo Sheet Properties

The physical organo sheet properties are calculated based on the measurements of weighing the stack before and the organo sheet after the process as well as thickness measurements at four points across the organo sheet surface, as depicted in Figure 8a. The measurements for these values allowed for the calculation of theoretical fiber and pore volume contents. Fiber volume content, $v_{f}$, was calculated using Equation (5), based on the textile stack weight determined before the process—each stack of three textile stacks has been weighed individually—with a density of 1.45 g/cm^3^ for hemp fibers. It was assumed that the fibers were completely dry—following the drying process in the convection oven—and that the reduction in the organo sheet weight can be attributed to polymer squeeze-out. The resulting volume was then divided by the area of the press tool with a diameter of 100 mm to arrive at a theoretical thickness. Three textile layers with an average of 10.5 g would thus have a theoretical thickness of 0.92 mm:



(5)
$$v_{f}=\frac{1}{t_{organo\;sheet}}\times \frac{m_{textile}}{\rho_{hemp}\times A_{tool}}$$



Matrix volume content $v_{m}$ was calculated in a similar manner, with (6) based on the total weight of the textile polymer stack, with the density of 1.26 g/cm^3^ for the recycled PLA based on the data sheet provided by the supplier:(6)$$v_{m}=\frac{1}{t_{organo\;sheet}}\times \frac{m_{polymer}}{\rho_{PLA}\times A_{tool}}$$

The pore volume content can finally be calculated with Equation (7) as the residual volume:(7)$$v_{p}=1-v_{f}-v_{m}$$

### 2.5. Evaluation of Organo Sheet Properties Based on Three Point Bending Tests

Three point bending tests, following DIN EN ISO 14125 [57], were used to investigate the effect of process parameters during organo sheet manufacturing. Specimens had a width of 15 mm and a length, l_part_, 20 times its height, t_part_. The support length L was set to 16 times the specimen thickness t_part_. The testing speed was set to 5 $\frac{mm}{min}$. Every test series produced five replicate organo sheets, with three specimens taken from each—a total of 15 specimens per parameter combination—for bending tests. Specimens were cut from the organo sheets with their length parallel to the warp direction, as depicted in Figure 8b. A schematic for the test setup for three point bending tests is shown in Figure 9.

## 3. Results

### 3.1. Polymer Properties

The specific polymer properties summarized in Table 1 were determined by using differential scanning calorimetry with a nitrogen atmosphere. The polymer was put into the chamber and heated to 210 °C (depicted in red), was then cooled down to 0 °C during the second cycle (depicted in black), before being heated once again to 210 °C (depicted in blue). Each step was conducted with a heating or cooling rate of 10 K/min. The heat flux curves for these steps allow different conclusions. The polymer shows limited crystallization behavior—the area above the reference line—during the first heating step, until the temperature is approximately 142 °C. The polymer can therefore be assumed to initially be in a crystalline state. The following temperature range is dominated by the resolution of crystalline structure—the area below the reference line—due to melting, with a peak at the location of the melt temperature at 154 °C. The curve arrives at a plateau once the polymer is molten completely, for temperatures higher than 167 °C. Temperatures for processing this polymer should preferably be higher than this threshold. Further heating to 210 °C shows no significant change in heat flux. The polymer behavior during cooling, below the glass transition temperature, was investigated during the second process step. The curve shows linear behavior between the maximum of 210 °C and the onset of glass transition at 62 °C. This leads to the conclusion that the polymer remains completely amorphous and does not crystallize during cooling down. The final process step was reheating the polymer to 210 °C. The first characteristic feature was the glass transition temperature, at temperatures virtually identical, at 59 °C, to the other two process steps. The second characteristic feature is the pronounced crystallization behavior, starting at approximately 100 °C. This area above the reference line is far more distinct during this second cycle than during the first heating cycle. The levels of the necessary energy for the formation and dissolution of the crystalline structure—striped areas below the curve—nearly cancel each other out, whereas the first heating cycle showed a larger negative value that can be attributed to the dissolution of the initial structure. The discussed value is described as “standardized” in Figure 10. The absence of crystallization during the cooling process can be attributed to the typically lower rate of crystallization of PLA when compared to other thermoplastics [42]. This might be a problem for the process evaluation based on the B-factor approach, since the recrystallization temperature constitutes the second temperature boundary. The reader is advised to compare this behavior to Figure A1, in which the DSC curve of a polypropylene bj100hp with distinct recrystallization behavior during cooling is depicted.

A comparison of the viscosities measured and predicted for the considered PLA is shown in Figure 11. The temperature-dependent values of viscosity can be calculated based on an Arrhenius equation; see Equation (8). In this equation, $\eta_{0}\left(T_{0}\right)$ is the selected viscosity at the reference temperature T_0_, $E_{a}$ is the activation energy, *R* is the universal gas and *T* is a given temperature within the range of interest:(8)$$\eta \left[T\left(t\right)\right]=\eta_{0}\left(T_{0}\right)\times \exp\left[\frac{E_{a}}{R}\times \left(\frac{1}{T}-\frac{1}{T_{0}}\right)\right]$$

Adequate agreement can be achieved, with the presented values for the parameters with T_0_ and η_0_ being taken from the data set and E_a_ being determined empirically.

### 3.2. Interpretation of Process Data and B-Factor Calculation

The process data for organo sheet manufacturing show characteristic behavior based on parameter combinations. Figure 12 shows a comparison of the change in stack thicknesses, as a result of different process pressures for a maximum process temperature of 170 °C. Curves for stack thickness change are displayed as the average and standard deviation of the five individual manufacturing processes. The effect of the process pressure is most pronounced between 0.5 MPa and 1.5 MPa, with a significant difference in thickness. The manifestation of this effect might be due to friction inside the yarn and fabric that is overcome when pressure is increased beyond 0.5 MPa. Compaction and nesting of the yarns and fibers within the textile structure might be the reason for an effect this pronounced. The difference between 1.5 MPa and 2.5 MPa is marginal and can only be identified in the early Sections (I and II) before the onset of macro-impregnation. This may be due to the stack being almost fully compacted and a limitted remaining compactibility, due to the nesting of the yarns of the textile and polymer stack, for pressures increasing beyond 1.5 MPa. The different stack thicknesses lead, in turn, to differences in heat conduction through the stack and a delay in the onset of macro-impregnation. The determined onset of macro-impregnation based on this data is highlighted as vertical lines for each pressure. The onset of macro-impregnation has been colorcoded to show the influence of process pressure.

The most distinct impact that can be identified is that increasing process pressure led to an earlier onset of macro-impregnation; see Figure 13a. This may be attributed to the pressure-dependent stack height. Air within the stack reduces heat conduction and thus delays the onset of macro-impregnation. An increasing pressure also led to smaller standard deviations. This delay caused different temperatures within the stack at the onset of macro-impregnation; see Figure 13b. The temperature within the stack converges with the melt temperature determined within DSC measurements, as indicated by the green line. The differences in temperatures at the onset of macro-impregnation can, on the other hand, be attributed to different heat rates for different target temperatures of the process. A delay in the onset of macro-impregnation means that different timespans—between 550 and 750 s—are available for the considered process, which has a significant impact on the overall impregnation; see Figure 13c. The individual temperature curves, as well as the different timespans, can be compared by calculating the B-factor for each parameter setting; see Figure 13d. The B-factors for these processes vary over a range of values between three and twenty that also show some overlap between different process temperatures and pressures. The values calculated for the processes of 170 °C, with a pressure of 2.5 MPa, and 185 °C, with a pressure of 1.5 MPa, are comparable with 12.2 and 13.4, respectively, and have overlapping standard deviations. The B-factor values for 185 °C with 2.5 MPa and 200 °C with 1.5 MPa allow for a similar comparison with 19.3 and 19.8, respectively. An overview of corresponding data for these figures is given in Table 3.

### 3.3. Examination of Physical Organo Sheet Properties

The chosen stack setup allows for instant investigation of the apparent impregnation quality, after the demolding of organo sheets. Molten polymer has to penetrate the stacks’ outer layers for proper impregnation, facilitating the ruling out of unsuitable process parameters for the given process time. Figure 14 shows a comparison of organo sheets, manufactured with different parameter combinations, one per test series, together with their respective B-factors. This comparison clearly indicates that certain parameter combinations are unsuitable for organo sheet manufacturing with the considered process time. For example, the low-pressure processes with maximum temperatures of 170 °C and 185 °C show dry spots of considerable size along the edge of the organo sheet. However, while the majority of the surface seems to be impregnated sufficiently, some dry areas can still be identified, for 170 °C and 1.5 MPa, along the edges. The evaluation of the surface quality might be an adequate tool for initial assessment, albeit a rather limited one. The figure therefore has to be considered in tandem with Figure 15, which shows an overview of the physical properties of the organo sheets that were determined after the manufacturing process. Figure 15a compares the influence of the different process parameters on the organo sheet thickness. The comparison to the target thickness—1.9 mm for a theoretical fiber volume content of 49%—should allow a first assessment of the organo sheets’ porosity. However, the resulting organo sheet weight has to be considered as well, to account for a reduced polymer volume content due to the squeeze-out of the molten polymer through the tool gap. Weighing the stack before and the organo sheet after the manufacturing process enabled consideration of this effect, as depicted in Figure 15b. The squeeze-out increased with both increasing process pressure and temperature. However, the effect of increasing temperatures was more pronounced, due to reductions in viscosity. This resulted in a polymer loss of 28% for the combination of 200 °C and 1.5 MPa. This polymer loss has an influence on the theoretical thickness of the organo sheet. A second comparison of the thicknesses measured for different parameter combinations and their theoretical counterparts is given in Figure 15c. The measured thicknesses are, in most cases, greater than the theoretical thicknesses, indicating that the excess volume is taken up by pores. The best agreement with minimal pores was determined to be the combination of 185 °C and 2.5 MPa, with an average pore volume content (PVC) of 0.16%. However, the corresponding standard deviation of 4.39% shows some limitations of this approach, overlapping into negative volumes. It is furthermore unlikely that natural fiber-reinforced composites made with thermoplastic polymers can be manufactured without pores. The fiber structure has a hollow lumen, and it is highly unlikely that the highly viscous polymer melt completely fills up this volume. However, the theoretical fiber and pore volume content can be calculated for each parameter combination, to enable a further means of comparison. The determined values are depicted in Figure 15d. The figure clearly shows decreasing PVC with an increasing pressure. The resulting FVC varies within the range between 37 and 51 vol. %. The effective FVC and PVC have to be considered when comparing mechanical properties in the following chapter. Parameter combinations with low pressures result in lower FVC near 37 vol. %, with a rather high PVC of 25 vol. %. Increasing both the pressure and the temperature facilitates a reduction in PVC, to an average of 6 vol. % at 1.5 MPa and 2 vol. % at 2.5 MPa, across all temperatures. A comprehensive overview of the data depicted in Figure 15 is given in Table A1.

### 3.4. Bending Properties

Bending moduli and strengths of the manufactured organo sheets, together with their fiber volume content, are presented in Figure 16. As discussed earlier, the process pressure in particular has a distinct effect on the fiber volume content of the organo sheets, resulting in two groups of values. The organo sheets manufactured with a pressure of 0.5 MPa displayed low bending moduli, with values barely competing with the unreinforced rPLA. Specimens for bending tests were cut from the area in the organo sheet where the polymer penetrated the top layers—predominantly dry areas were not tested. The values increase when pressure increases to 1.5 MPa and 2.5 MPa, especially with process temperatures of 170 °C. For temperatures of 185 °C, this effect is only comparable for pressures of 1.5 MPa, albeit less so than for 170 °C. A further increase to 2.5 MPa led to a reduction in the bending modulus to the level of the unreinforced rPLA. The effect of process temperature can be identified when comparing the organo sheets manufactured at 1.5 MPa highlighted in light yellow. Even though 170 °C is the lowest considered process temperature, it resulted in the highest values for this specific pressure level for bending moduli, and the highest overall bending strength with 51 ± 4 MPa. However, the overlap in standard deviations for 170 °C with 1.5 MPa and 2.5 MPa prevent the determination of significant differences. The reduction in both bending modulus and strength with an increasing temperature might be related to the oxidization and degradation of the matrix polymer in air [44,46,47,48]. This leads to drastically reduced bending moduli, at 2.1 GPa—60% of the raw material—and bending strengths as low as 20 MPa. Even though bending moduli showed a moderate reinforcement effect with elevated values that were 30% higher than the raw material, it was not possible to achieve a reinforcement effect when considering bending strengths. Similar effects have been reported in the literature and may be connected to limited fiber–matrix adhesion [36,37,38].

It was discussed earlier that especially low pressures lead to high pore volume content and, thus, a reduced mechanical performance. An additional set of five organo sheets was manufactured, with presumably the optimal process conditions of 170 °C and 1.5 MPa, to identify the effect of pore volume content. Additional polymer was added to the stack—17 g instead of the initial 9.3 g—to produce an organo sheet with a fiber volume content of 37%. The evaluation of stack thickness and polymer squeeze-out showed that the pore volume content could be reduced to 5%. Both bending modulus and strength increased, compared to the values determined in the first test series. A B-factor value of 8.7 ± 0.3 was determined for the corresponding process. The corresponding B-factor values for the process pressure and temperatures of 0.5 MPa and 170 °C, and 185 °C, were 3.1 ± 0.3 and 5.4 ± 0.1, respectively. This comparison is depicted in Figure 17. The colored arrows alongside the horizontal dashed lines illustrate the increase in bending modulus and strength that was achieved by application of the optimized process design.

The comparison of bending properties with the corresponding B-factor is presented in Figure 18. The effect of increasing B-factor on the bending moduli results in a parabolic course, with the zenith at a value of 12. A similar course can be described when bending strength is considered. The peak is slightly earlier, at a B-factor value of 10. The effect of temperature is distinct when comparing bending strengths at similar B-factors. This further highlights the necessity of considering the specific polymer behavior, even though the impregnation performance expressed by the B-factor might be of equal value. It might therefore be preferential to increase impregnation performance through increases in the impregnation time, for bio-based polymers with thermal sensitivity. The effect of increasing temperature (T↑) is highlighted in Figure 18b) for bending strengths with equal B-factors values.

The presented data show that the mechanical performance of the manufactured organo sheets decreases with increasing temperatures. However, the data do not allow the definite attribution of this behavior to degradation of the fiber or the matrix, since both PLA and hemp fibers may experience thermal degradation during the process. Therefore, an additional test series focused on manufacturing hemp fiber-reinforced organo sheets with a polypropylene matrix. The press processes were identical to those for hemp–rPLA organo sheets, with temperatures of 170 °C, 185 °C and 200 °C at a pressure of 1.5 MPa. An additional temperature level, at 215 °C, was added to investigate the effect of further increasing the process temperature on thermal fiber degradation. It can be assumed that the PP matrix does not experience thermal degradation within the considered temperature range, which allows us to draw conclusions regarding the degradation of fiber components. Figure 19 shows the temperature influence on bending moduli and strength for hemp–rPLA and hemp–PP organo sheets. As discussed earlier, hemp–rPLA organo sheets show a distinct reduction in bending properties with increasing temperatures. This behavior was not determined for hemp–PP organo sheets, which should allow the conclusion that the decrease in hemp–rPLA properties can be predominantly attributed to thermal degradation in matrix properties, rather than the damaging of natural fibers. An indicator for the occurring thermal fiber damage might be recognizable by the comparison of hemp–PP organo sheets manufactured at 200 °C and 215 °C. The average values for bending modulus and strength are slightly lower for 215 °C compared to 200 °C. Even though this effect can not be described as significant, due to an overlap in standard deviation, it might hint at a performance reduction due to degradation of lignin and pectin within the hemp fiber.

### 3.5. Consideration of Natural Fiber Morphology

As discussed in the Section 1, the morphology and properties of natural fibers may widely vary. This might be due to different factors, such as their positions within the plant stem—with decreasing diameter towards the top end—and climatic influences during plant growth and retting—especially considering the separation of technical fiber bundles. Figure 20 shows an excerpt of a microsection, taken from a hemp–rPLA organo sheet. The microsection was prepared on specimens after three point bending tests. Different areas within the microsection are highlighted to differentiate between fibers oriented in parallel (green) and perpendicular (blue) to the picture plane, areas of pure polymer (yellow) and the polyester twine used to stabilize the stable fiber yarn (purple). This twine shows a constant diameter in each fiber of approximately 20 microns and can therefore be used as a reference scale for comparison. Natural fibers oriented perpendicular to the plane (blue) show a high variance in both fiber diameters and shape. The figure also has some areas of interest outlined in white circles. These areas will be used to discuss challenges with regard to natural fibers as a continuous reinforcement structure. Area I focusses on clusters of fibers with moderate diameters ranging between 45 and 70 microns. The highlighted fibers show some separation lines, although it is not possible to distinguish if these are separate fibers or single fibers with cracks due to damaging. Hemp fibers with smaller diameters and distinct separation are also visible in the direct vicinity. Area II highlights two large clusters of fibers with a quasi-concentric arrangement. This could be a technical fiber bundle that is not separated completely, resulting in an elliptical cluster with a diameter between 180 and 200 microns for the inner and between 230 and 250 microns for the outer circle. Area III highlights a smaller cluster of elementary fibers with a diameter between 100 and 120 microns. Area IV highlights fibers with large diameters, similar to area I; however, the area is dominated by large fiber diameters rather than a mixture of large and small diameters. Finally, area V highlights an incompletely separated fiber bundle that resulted in an elongated ellipsis with an aspect ratio of six: one between the major and minor axes. These high variances might hinder exploitation of the full potential of natural fibers for structural applications.

## 4. Discussion

The production of organo sheets as a semi-finished product is a complex process with many influences. Process optimization is even more challenging when conventional petrol-based polymers and synthetic fibers are substituted with recycled and bio-based materials, like hemp fibers or recycled PLA. Investigating the melt behavior of recycled PLA in DSC showed that maximum process temperatures should at least be set to 170 °C to facilitate the total dissolution of crystalline structures. However, melt viscosity at this temperature is quite high, at 290 Pa∙s. The usual approach to reduce melt viscosity in thermoplastic polymers is to increase the temperature. This results in a drastic viscosity reduction to 95 Pa∙s at 185 °C and to 32 Pa∙s at 200 °C. This reduction in viscosity facilitates fiber impregnation, but also leads to an increasing polymer squeeze-out, of up to 30% of the initial polymer weight, through the tool gap. The presented results also show a distinct correlation between an increasing temperature and a decreasing performance in bending tests. A comparison of different matrix polymers showed that the reduced performance can be attributed to the thermal degradation of the recycled PLA, rather than the degradation of fiber components. It might be advantageous to increase impregnation performance based on the B-factor value by extending hold time at the maximum process temperature of 170 °C. This may allow for a higher degree of micro-impregnation and reduce the pore volume content below the average of 5%. The investigation of bending properties showed further limitations with regard to the exploitation of the properties of natural fibers as a reinforcement structure. While it was possible to achieve a reinforcement effect for a bending moduli of 39% compared to the raw polymer, it was not possible to increase the bending strength beyond a level of 50% of the raw material. This limitation with regard to natural fiber reinforcement of PLA is in accordance with the available literature and might be overcome by treatment of the fibers, to allow for a better fiber–matrix adhesion. The variance in fiber separation and diameter have to be considered, together with performance limitations such as the variance in mechanical properties due to environmental influences, as well as limited fiber–matrix adhesion. In particular, differences in diameter might limit the performance, for example, when smaller fibers tear at a lower force or fiber–matrix adhesion fails for large surface fibers.

## Figures and Tables

**Figure 1 polymers-15-04357-f001:**
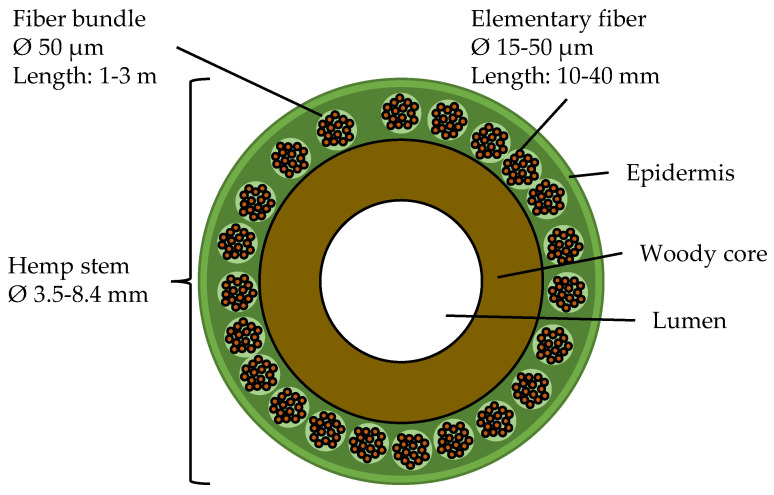
Schematic representation of the bast fiber structure, after [8,13].

**Figure 2 polymers-15-04357-f002:**
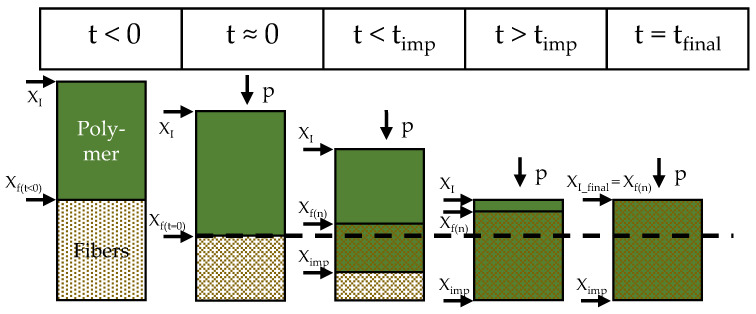
Schematic of impregnation process, based on [51] after [49,50].

**Figure 3 polymers-15-04357-f003:**
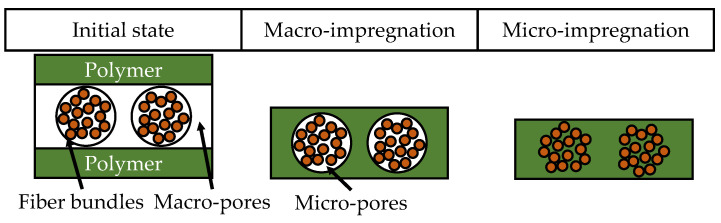
Schematic of macro- and micro-impregnation, after [51].

**Figure 4 polymers-15-04357-f004:**
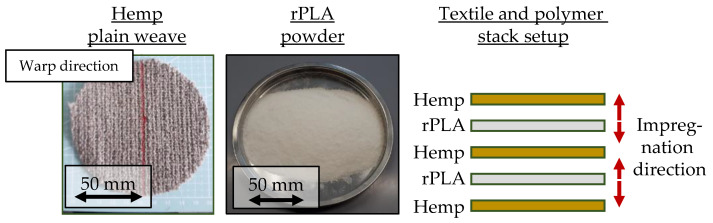
Dry textile (**left**), recycled PLA powder (**middle**) and stack setup (**right**).

**Figure 5 polymers-15-04357-f005:**
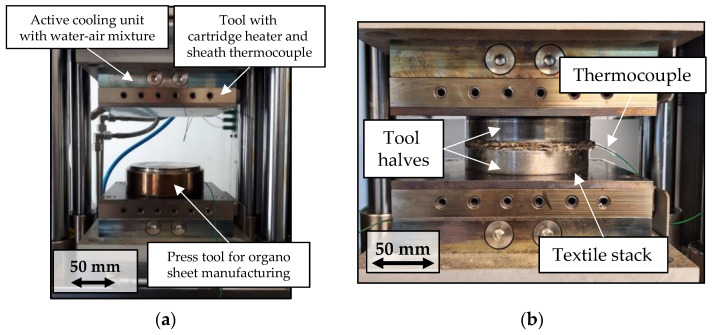
(**a**) Laboratory hot press with press tool; (**b**) setup for temperature measurement inside the fiber stack.

**Figure 6 polymers-15-04357-f006:**
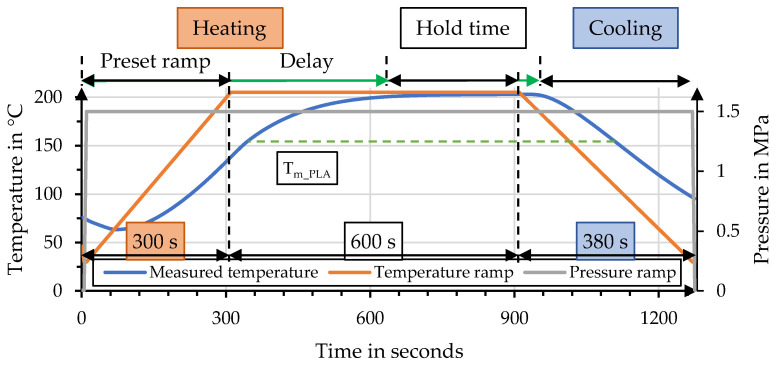
Comparison of temperature ramp and effective temperature alongside pressure ramp.

**Figure 7 polymers-15-04357-f007:**
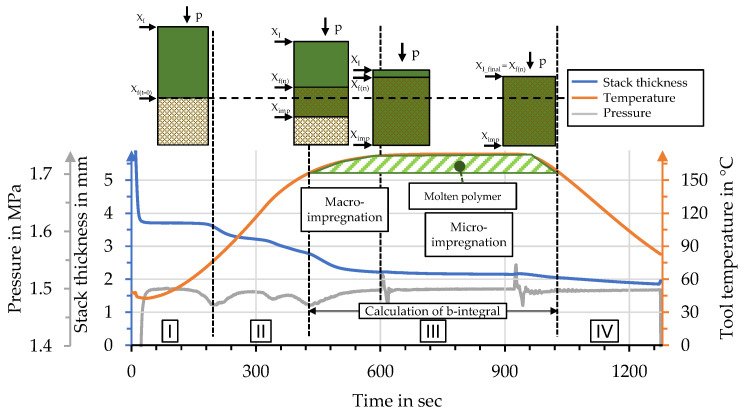
Correlation of process data with impregnation phases, with schematics based on [51] after [49,50].

**Figure 8 polymers-15-04357-f008:**
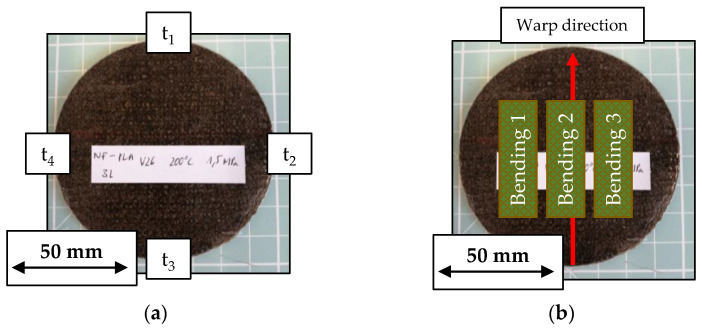
(**a**) Points for organo sheet thickness measurements and (**b**) position of three point bending specimens within the organo sheet.

**Figure 9 polymers-15-04357-f009:**
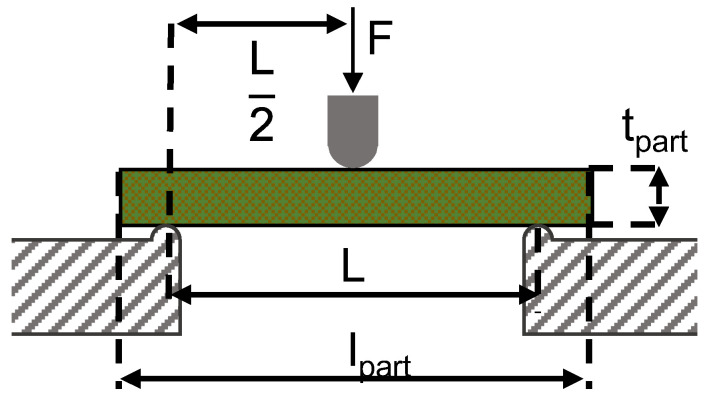
Setup for three point bending tests [57].

**Figure 10 polymers-15-04357-f010:**
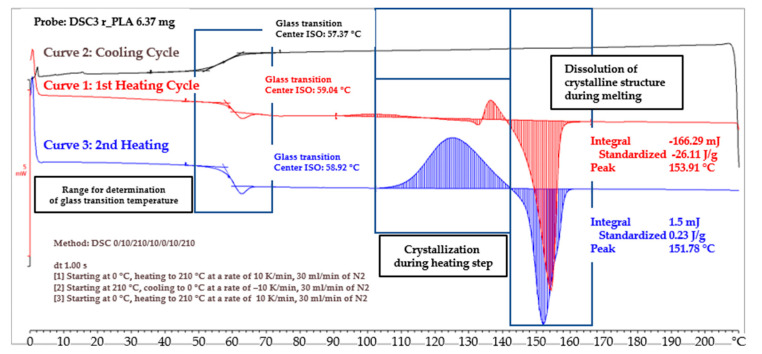
DSC curve recorded during differential scanning calorimetry.

**Figure 11 polymers-15-04357-f011:**
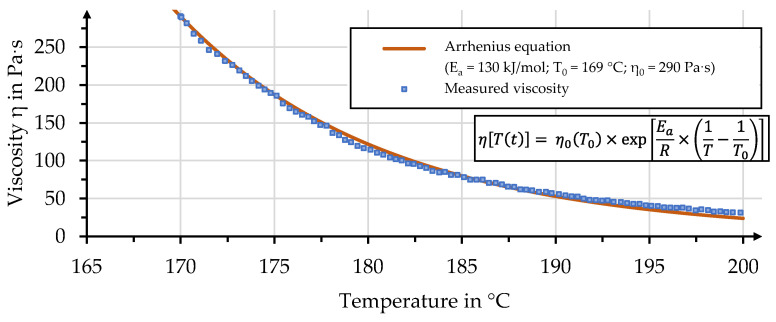
Comparison of measured and predicted viscosity for the considered rPLA based on an Arrhenius equation.

**Figure 12 polymers-15-04357-f012:**
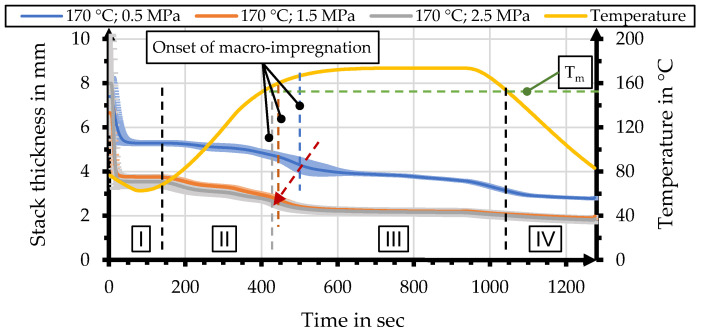
Comparison of stack thicknesses for processes with a maximum temperature of 170 °C and different process pressures.

**Figure 13 polymers-15-04357-f013:**
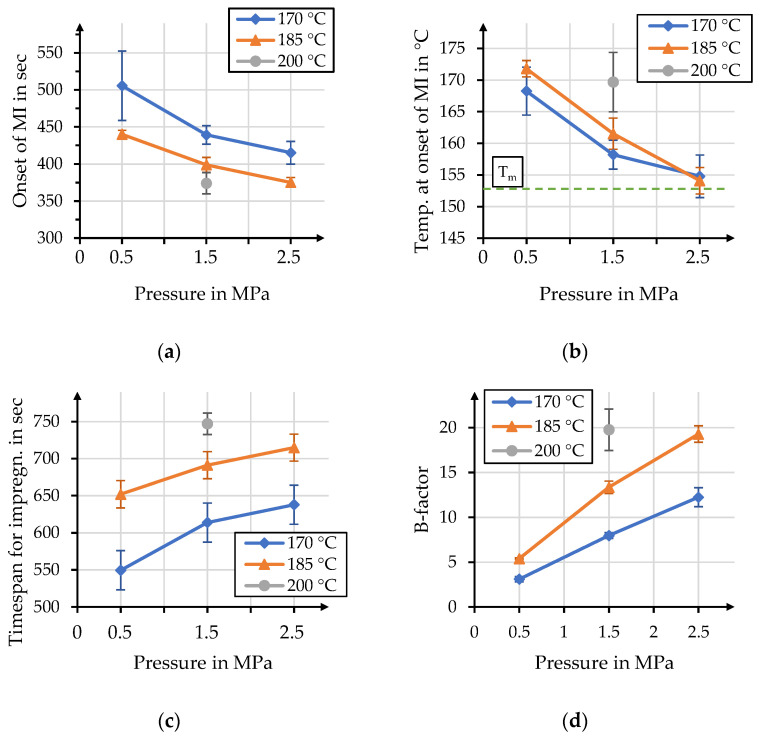
Influence of process pressure on the (**a**) timespan until the onset of macro-impregnation (MI); (**b**) stack temperature at the onset of macro-impregnation; (**c**) timespan for impregnation and (**d**) B-factor for each parameter combination.

**Figure 14 polymers-15-04357-f014:**
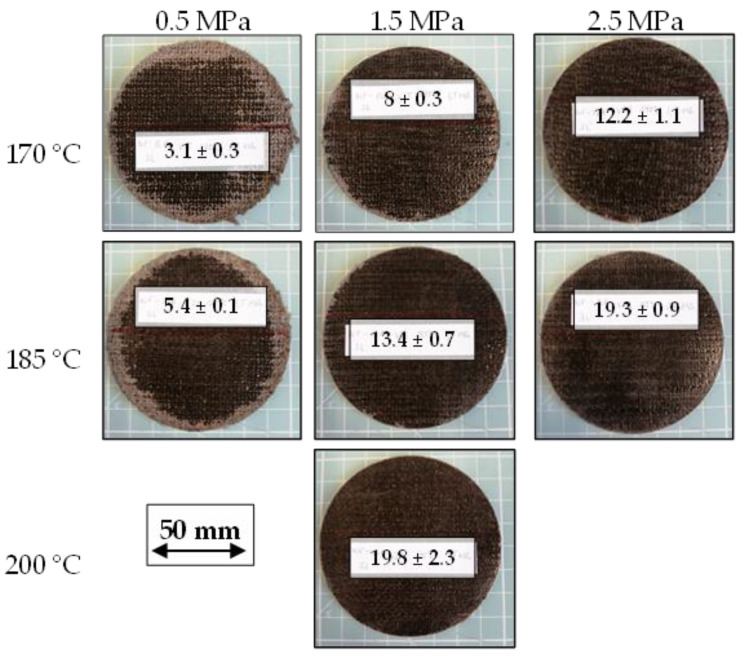
Comparison of apparent impregnation qualities with their B-factor.

**Figure 15 polymers-15-04357-f015:**
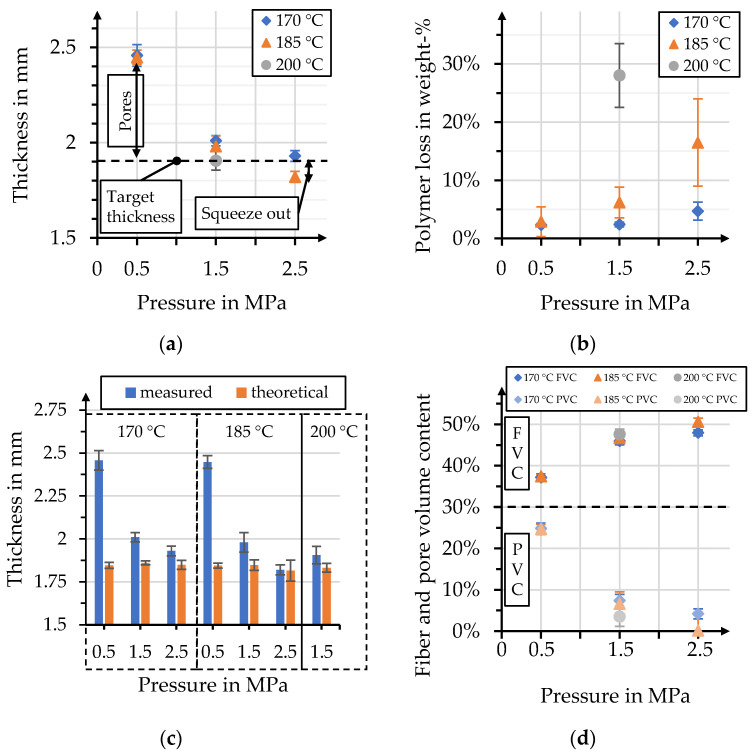
(**a**) Measured organo sheet thickness in comparison with target thickness; (**b**) polymer loss due to squeeze-out through the tool gap; (**c**) comparison of measured and theoretical organo sheet thickness based on its weight; (**d**) theoretical fiber and pore volume content based on organo sheet weight and thickness.

**Figure 16 polymers-15-04357-f016:**
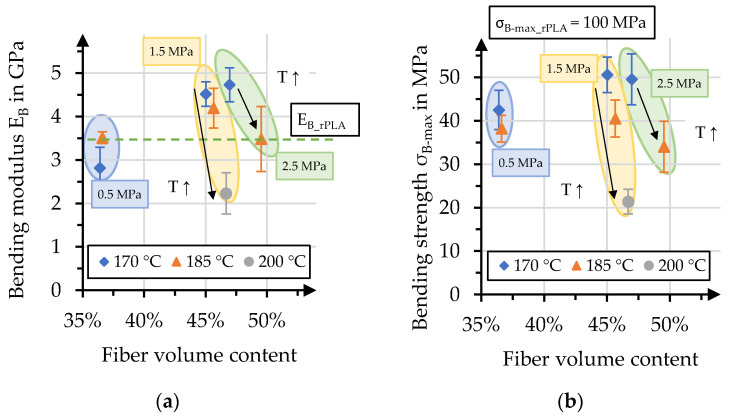
(**a**) Bending moduli and (**b**) bending strengths, compared to effective fiber volume content.

**Figure 17 polymers-15-04357-f017:**
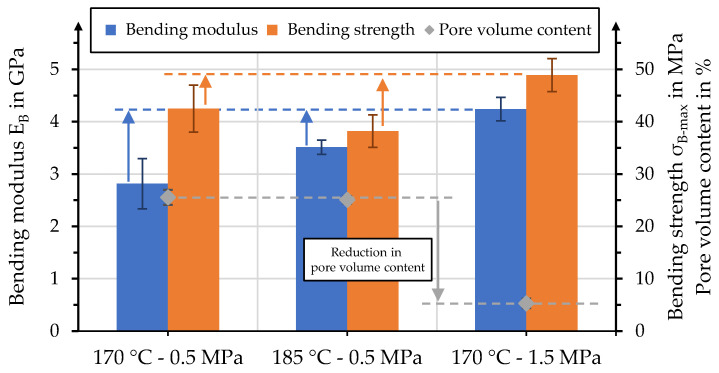
Effect of pore volume content on bending properties for a FVC of 37%.

**Figure 18 polymers-15-04357-f018:**
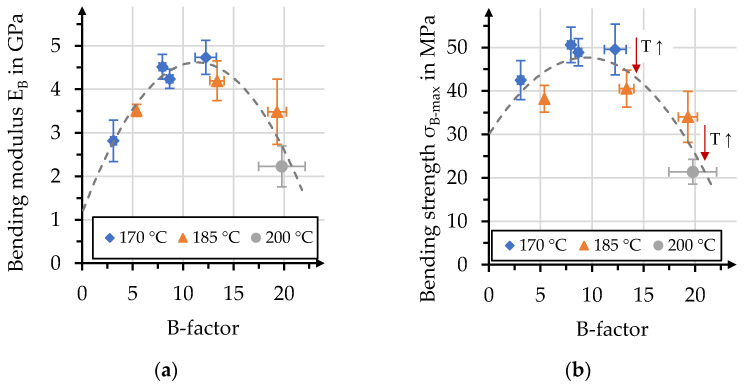
(**a**) Bending modulus and (**b**) bending strengths compared to B-factors for each process.

**Figure 19 polymers-15-04357-f019:**
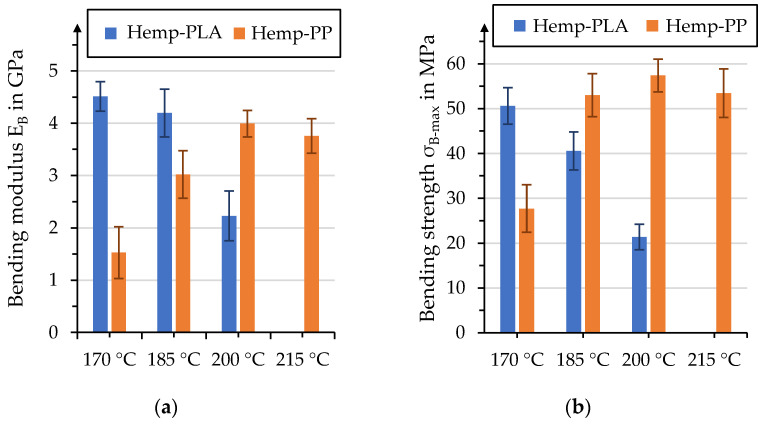
Comparison of temperature influence in hemp–rPLA and hemp–PP organo sheets on (**a**) bending modulus and (**b**) bending strength.

**Figure 20 polymers-15-04357-f020:**
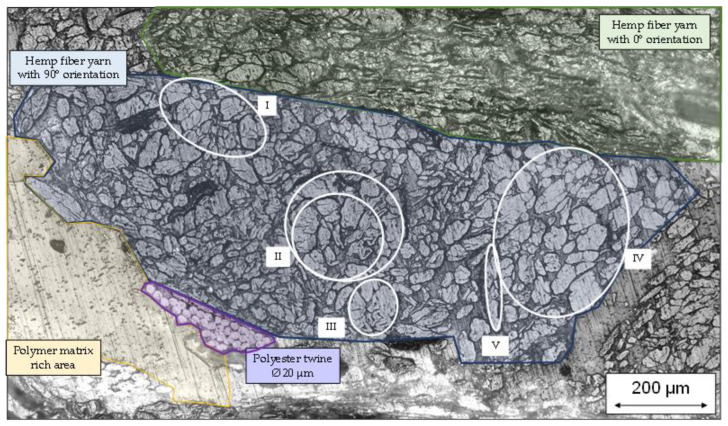
Microsection of hemp–rPLA organo sheet.

**Table 1 polymers-15-04357-t001:** Polymer properties of recycled PLA and PP.

Polymer	Density	Glass Transition Temperature	Melt Temperature	Bending Modulus	Bending Strength
rPLA (Looplife polymers)	1.26 g/cm^3^	59 °C	152.8 °C	3.44 GPa	98 MPa
PP (Borealis bj100hp)	0.9 g/cm^3^	<0 °C *	165 °C	1.25 GPa	35 MPa

* Glass transition temperature of polypropylene bj100hp was not detected in DSC between 0 °C and 200 °C.

**Table 2 polymers-15-04357-t002:** Process parameter for manufacturing of hemp fiber-reinforced organo sheets.

Material	Temperature	Pressure	Fiber Volume Content
Hemp–rPLA	170 °C 185 °C 200 °C	0.5 MPa 1.5 MPa 2.5 MPa	37–50 vol.-%
Hemp–PP	170 °C 185 °C 200 °C 215 °C	1.5 MPa	43–48 vol.-%

**Table 3 polymers-15-04357-t003:** Comparison of characteristic values determined for different parameter settings.

Temperature	Pressure	Onset of MI in Seconds	Temp. at Onset	Timespan for Impregnation in Seconds	B-Factor
170 °C	0.5 MPa	506 ± 46	168 ± 4 °C	550 ± 46	3.1 ± 0.3
170 °C	1.5 MPa	439 ± 12	158 ± 2 °C	614 ± 12	8 ± 0.3
170 °C	2.5 MPa	415 ± 15	155 ± 3 °C	638 ± 15	12.2 ± 1.1
185 °C	0.5 MPa	440 ± 6	172 ± 1 °C	652 ± 6	5.4 ± 0.1
185 °C	1.5 MPa	399 ± 10	162 ± 3 °C	691 ± 10	13.4 ± 0.7
185 °C	2.5 MPa	375 ± 7	154 ± 2 °C	715 ± 7	19.3 ± 0.9
200 °C	1.5 MPa	374 ± 15	165 ± 5 °C	747 ± 15	19.8 ± 2.3

## Data Availability

The data presented in this study are available on request from the corresponding author.

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
