# Peer review of "Challenges in Manufacturing of Hemp Fiber-Reinforced Organo Sheets with a Recycled PLA Matrix"

_polymers, 2023, doi:10.3390/polym15224357_

Round 1

Reviewer 1 Report

Comments and Suggestions for Authors

The present paper titled “Challenges in manufacturing of hemp fiber reinforced organo sheets with a recycled PLA matrix” is focused on the production and mechanical properties of hemp/PLA composite materials and on the investigation of the process parameters in terms of pressure, temperature and hold time during a hot compression moulding production technology. A previous analysis on the glass transition and on the melting temperature have been performed by means of a differential scanning calorimetry (DSC) aiming to have a better insight into the suitable process parameters of the used matrix. A defined value of the fibre’s impregnation was evaluated with the B-factor model using the process parameters and the polymer’s viscosity at the processing temperature. Then, three-point bending tests were performed on each sample typology, and the influence of the process parameters on the mechanical properties was evaluated.

Therefore, an overview on the research work lets to conclude that, even if the issues about the thermal stability of natural fibres and PLA matrix and the adhesion efficiency at the fibre-matrix interface, the present paper can be considered of interest for the scientific community. However, it requires some minor revision before the publication.

-          Introduction

This section is well written, the authors have meticulously described natural fibres with an insight into the structure of this category of reinforcement and described the thermal degradation process that affect the PLA matrix. A detailed description of the dry fabric reinforcement impregnation was presented, and a satisfying B-factor model definition was performed. However, the reviewer suggests a reduction of some sentences and recommends an improve of the theoretical background on the hot compression moulding of natural fibres and the issue about the adhesion efficiency at the interface between matrix and natural fibres.

-          Materials and Methods

1)      Page 5, lines 180 – 189. A repetition was found by the reviewer, the sentence “Glass transition and melt temperature of the polymers have been determined by…. Viscosity measurements have been conducted for 10 % 188 strain with a frequency of 10 Hz.” Is repeated in the same page from line 166 to 175. Please check and rearrange the main text.

2)      Page 6, lines 206 – 209. It is clear that due to the hydrophilic behaviour, natural fibres are prone to absorb moisture, therefore a drying step is mandatory before the fibre impregnation. However, this aspect is not clear, therefore, please rearrange the sentence.

3)      Page 6, lines 222 – 226. Please rearrange the list referred to the compression moulding apparatus.

4)      Page 9 line 303. The standard used for the bending test usually recommends a support span that is function of the sample’s thickness, however, in the manuscript the authors set the span as function of the sample’s length. Is it a typing error? If yes, please rearrange the section.

-          Results

1)      Page 10, lines 315 – 324.  The sentence “the specific polymer properties summarized….strain with a frequency of 10 Hz.” was repeated in the Materials and Methods section, therefore, please rearrange the manuscript. If the authors are in accordance with the reviewer, I suggest to place the sentence in the Materials and Methods section.

2)      Page 11, line 355. The axis labels in figure 10 are too small and not easy to read, therefore, please increase the font size.

3)      Page 12, lines371 – 377. The reviewer agrees with the authors about the effect of the process pressure passing from 0.5 MPa to 1.5 MPa since the increase in the pressure leads to a significant fibre’s and polymer compaction, then to a better heat transfer and then to a better fibre impregnation. However, can the authors have a better insight and better explain why passing from 1.5 MPa to 2.5 MPa the difference in the produced samples is marginal?

4)      Page 16, lines 466 – 477. The authors pointed out that using a process pressure of 2.5 MPa and a temperature of 200 °C, it is possible to obtain a high level of fibre impregnation since these process parameters allows to the highest B-factor. The data shows a high fibre volume percentage and a reduced presence of residual porosities therefore, high mechanical properties are expected from the produced sample. Furthermore, at line 476 the authors sentenced that the high temperature leads to the polymer oxidation, and therefore to a reduction of its mechanical properties. However, at this temperature, hemp fibres are well impregnated due to the reduced matrix viscosity, therefore, since the reinforcement has load bearing properties, it is expected that the mechanical properties of the overall composite are not significantly compromised. Therefore, based on these observations, can the authors further justify the obtained results?

5)      Page 16, lines 493 – 494. What is the previous value of the B-factor? Please specify in the revised manuscript.

6)      Page 17, lines 510 – 512. Please revise the sentence.

Comments on the Quality of English Language

the paper is written using a good English language

Reviewer 2 Report

Comments and Suggestions for Authors

The author should incorporate the following suggestions/comments in their manuscript.

1. In The introduction section, the author should add more references from 2022 and 2023. It is very hard to find a clear research gap. Further, the objective of the work should be elaborated more for more clarity.

2. The supplier name should be added in the materials and methods section for materials. Further, the model and make of the equipment for carried out analysis.

3. The author should discuss the crystallization effect in DSC analysis in a proper scientific manner

4. The author should write the Arrhenius equation for better clarity.

5. The effect of the process pressure is most pronounced between 0.5 MPa and 1.5 MPa with a significant difference in thickness. Justify

6. The discussion section should be elaborated more with respect to the obtained results.

7. The author should add the conclusions of their work. It is missing.

Round 2

Reviewer 2 Report

Comments and Suggestions for Authors

No Comments